

# Comparison of frozen-thawed embryo transfer strategies for the treatment of infertility in young women: a retrospective study

Yanhong Wu[1], Xiaosheng Lu[1], Yanghua Fu[1], Junzhao Zhao[1] and Liangliang Ma[2]

[1] Department of Obstetrics and Gynecology, The Second Affiliated Hospital and Yuying Children's Hospital of Wenzhou Medical University, Wenzhou, Zhejiang, China
[2] Department of Vascular Surgery, The Second Affiliated Hospital and Yuying Children's Hospital of Wenzhou Medical University, Wenzhou, Zhejiang, China

Corresponding authors
Junzhao Zhao, z.joyce08@163.com
Liangliang Ma, 330592479@qq.com

## ABSTRACT

**Objective**. To investigate transfer strategies in the frozen-thawed embryo transfer (FET) cycle.

**Methods**. The clinical data of 1,652 FET patients were divided into five groups according to the number and quality of the transferred blastocyst: high-quality single blastocyst group (group A, $n = 558$), high-quality plus poor-quality double blastocyst group (group B, $n = 435$), poor-quality double blastocyst group (group C, $n = 241$), high-quality double blastocyst group (group D, $n = 298$), and poor-quality single blastocyst group (group E, $n = 120$). Inter-group comparison analyses of primary conditions, pregnancy outcomes and neonatal outcomes were then performed.

**Results**. Group A had the highest embryo implantation rate (67.38%), significantly different from the implantation rates of the other four groups. The gemellary pregnancy rate (1.60%), preterm birth rate (5.58%), neonatal birth weight (3,350g [3,000g, 3,650g]), neonatal birth age (39.57 weeks [38.71, 40.34]), and incidence of low birth weight (7.02%) in group A were different from those in groups B, C, and D, but did not significantly differ from those in group E. Moreover, the proportions of male infants born in groups A (56.86%) and D (59.41%) were significantly higher than those in the other three groups. Double blastocyst transfer (0.528, 95% CI [0.410–0.680], $P < 0.001$) and high-quality blastocyst transfer (0.609, 95% CI [0.453–0.820], $P = 0.001$) were found to be protective factors for live birth. In addition, double blastocyst transfer was also the largest risk factor for pregnancy complications (3.120, 95% CI [2.323–4.190], $P < 0.001$) and neonatal complications (2.230, 95% CI [1.515–3.280], $P < 0.001$), especially for gemellary pregnancy (59.933, 95% CI [27.298–131.58], $P < 0.001$) and preterm birth (3.840, 95% CI [2.272–6.489], $P < 0.001$). Based on the ROC curves, a double blastocyst transfer could predict gemellary pregnancy reliably with a high area under the curve (AUC = 78.53%). Additionally, a double blastocyst transfer could effectively predict a high risk of pregnancy complications (AUC = 65.90%), neonatal complications (AUC = 64.80%) and preterm birth (AUC = 66.20%).

**Conclusion**. The live birth rate of frozen-thawed high-quality single blastocyst transfer is lower than that of double high-quality blastocyst transfer, which can significantly increase the embryo implantation rate. High-quality single blastocyst transfer also significantly lowers the risk of gemellary pregnancy, preterm birth, and low birth

weight, and can significantly improve maternal and infant outcomes. After weighing the pros and cons of live birth with pregnancy and neonatal complications, the authors believe that high-quality single blastocyst transfer is the optimal FET strategy for young women and is worthy of further clinical application. Despite this recommendation, high-quality single blastocyst transfer can increase the risk of monozygotic twins, as well as significantly increase the proportion of male infants born.

## INTRODUCTION

Assisted reproductive technology (ART) has been around for over 40 years. The blastocyst culture and transfer technique is a widely used technique that improves the pregnancy outcomes of ART (*Freeman et al., 2019*). As blastocyst transfer rates have increased, so have ART pregnancy rates including gemellary pregnancy rate. An important reproductive medicine research topic in recent years has been: maintaining high pregnancy rates and good pregnancy outcomes with ART while reducing ART gemellary pregnancy rate. Several studies show that single embryo transfer (SET) has a clinical pregnancy rate similar to double embryo transfer (DET), and is the most effective way to reduce the risk of gemellary pregnancy with ART (*Cutting, 2018*; *Racca et al., 2020*).

In 2009, the British Human Fertilization and Embryology Society (HFEA) issued a policy requiring the routine application of SET in reproductive centers. Since then, the ART gemellary pregnancy rate in the UK has dropped from 26.6% in 2008 to 16.3% in 2013, and the overall live birth rate with ART has not been affected (*Harbottle et al., 2015*). Over the past decade, in order to reduce the ART gemellary pregnancy rate, European, American, Japanese, and Australian scientists have actively promoted SET, achieving a SET rate of 50%–85% (*De Geyter et al., 2020*; *Kushnir et al., 2017*; *Dyer et al., 2016*). ART in China has undergone a dramatic development in the past 30 years, but the gemellary pregnancy rate still remains high. According to the CSRM Assisted Reproductive Technology Data Reporting System of the Chinese Medical Association Reproductive Medicine Branch, China's 2020 fresh cycle and frozen-thaw cycle gemellary pregnancy rates are as high as 25.94% and 20.68%, respectively, while the gemellary pregnancy rate in the United States was only 14.7% in 2017 (*Sunderam et al., 2020*). Although Chinese experts achieved a consensus on promoting SET in 2018, due to a lack of knowledge of SET and the preference for twins in China, implementing SET in China has been difficult. Many also have concerns about SET, including a fear that SET will reduce the live birth rate, especially in the frozen-thawed embryo transfer (FET) cycle, and the scope of current SET implementation in China is unknown.

This study retrospectively analyzed the clinical data of 1,652 patients who received FET in the reproductive center of the Second Affiliated Hospital of Wenzhou Medical University from January 2019 to December 2020 and analyzed the pregnancy and neonatal outcomes

of five FET strategies. The conclusions of this study can help provide a reference for the optimal FET strategy for treating infertility in young women.

## MATERIALS & METHODS

### Research objects

A retrospective analysis of females undergoing FET was performed from January 2019 to December 2020 in the reproductive center of the Second Affiliated Hospital of Wenzhou Medical University. The inclusion criteria were as follows: (1) all patients were between the ages of 20 and 35 years old, and the causes of infertility included tubal obstruction, endometriosis (EMs), polycystic ovary syndrome (PCOS), male factors, or unexplained reasons; (2) the endometrial layer thickness was greater than seven mm on the day of endometrial transformation; (3) patients had no more than two transplant cycles; (4) single or double Day 5 (D5) blastocysts were transferred; (5) hormone replacement therapy was used for endometrial preparation. The exclusion criteria were as follows: (1) Uterine abnormality under ultrasound such as endometrial polyps, endometrial fibroids, intrauterine adhesion or uterine malformation; (2) malignant tumor or other systemic diseases; (3) congenital genetic abnormalities; (4) genital tract malformations; (5) history of recurrent miscarriage; (6) Day 6/7 (D6/7) blastocyst or Day 3/4 (D3/4) cleavage stage embryo transferred; (7) the use of other endometrial preparation programs, such as daily natural therapy, ovulation induction therapy, and gonadotrophin releasing hormone analogue (GnRH-a) down-regulating hormone replacement therapy.

A total of 1,652 eligible patients were included in the study and divided into five groups according to the number and quality of transferred blastocysts: high-quality single blastocyst group (group A, $n = 558$), high-quality plus poor-quality double blastocyst group (group B, $n = 435$), poor-quality double blastocyst group (group C, $n = 241$), high-quality double blastocyst group (group D, $n = 298$), and poor-quality single blastocyst group (group E, $n = 120$; Fig. 1).

This study was approved by the Ethics Committee (Institutional Review Board) of the Second Affiliated Hospital and Yuying Children's Hospital of Wenzhou Medical University and informed written consent was obtained from all participants (2021-K-74-02).

### Frozen-thawed single blastocyst transfer cycle

HRT patients commenced oral administration of one tablet of estradiol (Femoston; Abbott Biologicals B.V. Dose 2 mg estradiol/tablet) twice a day on the 2nd-5th day of the menstrual cycle. A B-ultrasound was performed every 3 to 5 days to measure the endometrial thickness and adjust the estradiol dosage accordingly. When the endometrial thickness ≥8 mm and serum progesterone levels <1.5 ng/ml, patients were given 10 mg dydrogesterone tablets (Duphaston; Solvay Pharmaceuticals B,V. Dose 10 mg/tablet), in addition to the one tablet of estradiol, orally twice a day and 200 mg micronized progesterone was also administered orally or vaginally twice a day (Utrogestan; Capsugel, Besins Manufacturing Belgium, Bruxelles, Belgium. Dose: 0.1 g/tablet). Single or double blastocysts for embryo transfer were selected on the 5th day after the endometrial transfer. The methods used for luteal phase support were the same as those used after endometrial transformation (Fig. 2).

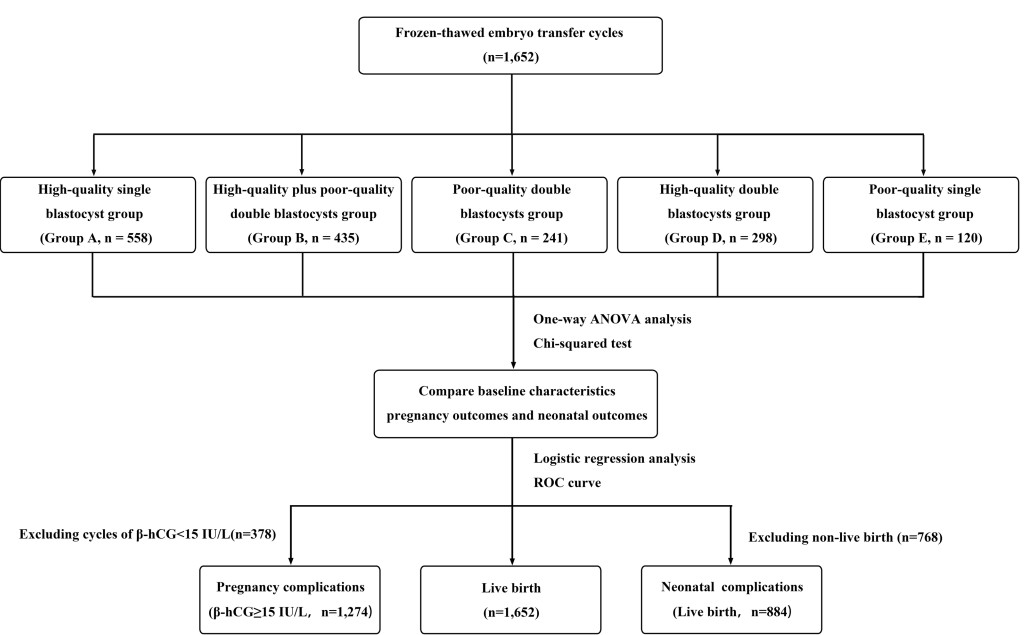

**Figure 1  Flow chart.** A total of 1,652 cycles received frozen-thawed embryo transfer (FET) in the reproductive center of the Second Affiliated Hospital of Wenzhou Medical University from January 2019 to December 2020. Among them, according to the number and quality of transferred blastocysts, the patients were divided into five groups: high-quality single blastocyst group (group A, $n = 558$), high-quality plus poor-quality double blastocyst group (group B, $n = 435$), poor-quality double blastocyst group (group C, $n = 241$), high-quality double blastocyst group (group D, $n = 298$) and poor-quality single blastocyst group (group C, $n = 241$). Statistical analysis was used to compare patients' data.

**Figure 2  Hormone replacement therapy.** Patients commenced oral estradiol on the 2nd-5th day of the menstrual cycle. A B-ultrasound was performed every 3 to 5 days to measure the endometrial thickness and to adjust the dosage of the estradiol accordingly. When the endometrial thickness ≥ 8 mm and serum progesterone levels <1.5ng/ml, progesterone would be given to start the endometrial transformation. Then, single or double blastocysts for embryo transfer were selected on the 5th day after the endometrial transfer. The methods used for luteal phase support were the same as those used after endometrial transformation. A $\beta$-hCG test was performed on day 12 of the FET cycle. The first B-ultrasound was performed after the 26th day of the FET cycle and the second B-ultrasound was on day 40.

## Thawing and culturing frozen-thawed embryos

On the morning of transplantation, the blastocysts were thawed using a vitrification recovery kit (Vitrification VT102; Kitazato, Shizuoka, Japan). The carrier rod was removed

from the liquid nitrogen, and then the casing was taken out and quickly placed in the thawing solution (TS) at room temperature for 1 min. The blastocysts were then transferred into the diluent (DS) for 3 min, washing solution 1 (WS1) for 5 min, washing solution 2 (WS2) for 5 min, and finally transferred to a pre-prepared petri dish, washed three times and left to culture until transplantation.

## Criteria for blastocyst evaluation

After the blastocysts were thawed, they were graded according to the Gardner classification system (*Gardner et al., 2000*), which includes the following metrics: Score the expansion stage and incubation status of blastocysts with number 1-6. Score the inner cell mass (ICM) and trophectoderm (TE) with letter A, B or C. Blastocysts with a development stage higher than 2, ICM grade higher than C, and TE grade higher than C (≥3BB) were considered high-quality, and poor-quality blastocysts were considered with scores lower than 3BB (*Gardner et al., 2004*; *Oron et al., 2014*).

## Follow-Up

A $\beta$-hCG test was performed on day 12 of the FET cycle. The first B-ultrasound was performed after the 26th day of the FET cycle and the second B-ultrasound was on day 40 (Fig. 2). A serum hCG ≥15 mlU/mL and no gestational sac 45 days after embryo transfer was defined as a biochemical pregnancy. Clinical pregnancy was identified as the presence of gestational sac in utero and fetal heart beat under transvaginal ultrasound. Early miscarriage was defined as fetal loss before 12 gestational weeks, while late miscarriage was defined as fetal loss after 12 gestational weeks. Preterm birth occurred between 28 to 37 gestational weeks. Infants with a neonatal birth weight <2500 g were considered low birth weight infants, while a neonatal birth weight ≥4000 g defined fetal macrosomia.

Pregnancy complications included in this study were: biochemical pregnancy, miscarriage, ectopic pregnancy, gemellary pregnancy, gestational diabetes mellitus (GDM), and intrahepatic cholestasis (ICP). Neonatal complications included: preterm birth, low birth weight, birth defect, and macrosomia.

## Statistical methods

Continuous variables are presented as the mean and standard deviation or median, and the comparison between the five groups was performed using one-way analysis of variance. In contrast, data not conforming to normal distribution was presented as medians and interquartile ranges (IQRs), and five comparisons between groups were performed using the Kruskal-Wallis H test. Clinical variables were compared using independent $t$-test, Pearson's chi-square test or Mann–Whitney U test, and the Fisher's exact test as appropriate. Multivariate logistic regression was used to analyze major factors affecting live birth, pregnancy complications and neonatal complications, and odds ratios (ORs) and 95% confidence intervals for independent variables were calculated. Receiver operating characteristic (ROC) curves were used to assess whether double embryo transfer and high-quality blastocyst transfer could be used to accurately predict live birth, pregnancy complications, gemellary pregnancy, neonatal complications, and preterm birth. All

analyses were two-tailed and difference of $P < 0.05$ was considered as statistically significant. All data were analyzed using SPSS (version 26.0; IBM, Chicago) in this study.

## RESULTS

### Comparison of baseline characteristics

There were no significant differences in maternal age, infertility duration, maternal BMI, infertility type, basal endocrine levels, and endometrial thickness on transfer day among the five groups (all $P > 0.05$). Group A was statistically different from group C in the amount of infertility cases that were caused by male factors (9.68% *vs.* 3.73%, $P = 0.004$) or that were unexplained (11.29% *vs.* 18.27%, $P = 0.005$). Both factors in group D were statistically different from group A (28.52% *vs.* 20.97%, $P = 0.013$) and group C (28.52% *vs.* 20.75%, $P = 0.038$). The proportion of patients in the first transplant cycle was highest in group A, which was significantly different from the other four groups (71.68%, 36.09%, 34.85%, 47.31%, 51.67%, respectively; all $P < 0.001$), and the proportion of patients in the first cycle was the lowest in group C, which was significantly different from group D (34.85% *vs.* 47.31%, $P = 0.004$) and group E (34.85% *vs.* 51.67%, $P = 0.002$; Table 1).

### Comparison of pregnancy outcomes and neonatal outcomes

There were no significant differences in the miscarriage rate, ectopic pregnancy rate, birth defect rate, or rate of obstetric complications among the five groups (all $P > 0.05$). The rate of positive hCG tests (76.52%, 81.38%, 77.18%, 81.21%, 54.17%, respectively; all $P < 0.001$) and the clinical pregnancy rate (67.38%, 68.74%, 60.58%, 73.15%, 37.50%, respectively; all $P < 0.001$) in group E were significantly lower than those of the other four groups. Group A had the highest embryo implantation rate, which significantly differed from the other four groups (67.38%, 49.77%, 41.70%, 55.20%, 37.50%, respectively; all $P < 0.001$). The gemellary pregnancy rate (1.60%, 44.82%, 37.67%, 50.92%, respectively; all $P < 0.001$) and the preterm birth rate (5.58%, 17.06%, 17.13%, 19.26%, respectively; all $P < 0.001$) were the lowest in group A compared to groups B, C, and D, there was no statistical difference between group A and group E ($P > 0.05$). The biochemical pregnancy rate in group A (9.14%) was significantly different from groups C (16.60%; $P = 0.002$) and E (16.67%; $P = 0.015$). Group D had the highest live birth rate (62.75%), which differed significantly from group A (52.51%; $P = 0.004$), group C (48.55%; $P < 0.001$), and group E (26.67%; $P < 0.001$). In group A, neonatal birth weight was significantly higher (3,350g [3,000g, 3,650g], 2,750g [2,350g, 3,300g], 2,950g [2,467g, 3,300g], 2,750g [2,450g, 3,300g], respectively; all $P < 0.001$), neonatal birth age was significantly higher (39.57 weeks [38.71, 40.34], 38.86 weeks [37.14, 40.00], 38.43 weeks [37.07, 39.50], 38.29 weeks [37.14, 39.16], respectively; all $P < 0.001$), and the incidence of low birth weight infants was significantly lower (7.02%, 27.78%, 26.32%, 27.68%, respectively; all $P < 0.001$) than groups B, C, and D (but were not statistically different than these values in group E). The incidence of macrosomia in group E (15.63%) was significantly higher than in group B (4.68%, $P = 0.010$), group C (3.95%, $P = 0.011$), and group D (2.95%, $P = 0.001$), but did not significantly differ from group A. The proportions of male infants born in groups A (56.85%) was significantly higher than the proportions of male infants born in groups

Wu et al. (2022), *PeerJ*, DOI 10.7717/peerj.14424

**Table 1  Comparison of baseline characteristics.**

| | Group A (n = 558) | Group B (n = 435) | Group C (n = 241) | Group D (n = 298) | Group E (n = 120) | P value |
|---|---|---|---|---|---|---|
| Maternal age, mean (SD) (year) | 29.97 ± 3.20 | 30.04 ± 3.10 | 30.17 ± 3.30 | 30.02 ± 3.34 | 30.11 ± 3.39 | 0.946 |
| Infertility duration, mean (SD) (year) | 3.25 ± 2.25 | 3.26 ± 2.12 | 3.07 ± 2.13 | 3.13 ± 2.27 | 3.44 ± 2.13 | 0.554 |
| Maternal BMI, mean (SD) (kg/m$^2$) | 21.56 ± 3.19 | 21.56 ± 3.01 | 21.95 ± 2.99 | 21.78 ± 3.13 | 22.00 ± 3.38 | 0.307 |
| Infertility type | | | | | | |
|     Primary infertility % (n) | 46.95(262/558) | 43.45(189/435) | 49.79(120/241) | 45.30(135/298) | 42.50(51/120) | 0.498 |
|     Secondary infertility % (n) | 53.05(296/558) | 56.55(246/435) | 50.21(121/241) | 54.70(163/298) | 57.50(69/120) | 0.498 |
| Infertile causes | | | | | | |
|     Female factor % (n) | 58.06(324/558) | 57.24(249/435) | 56.85(137/241) | 52.35(156/298) | 58.33(70/120) | 0.574 |
|     Male factor % (n) | 9.68(54/558)[b] | 6.67(29/435) | 3.73(9/241) | 6.04(18/298) | 7.50(9/120) | 0.036* |
|     Both factors % (n)[#] | 20.97(117/558)[c] | 22.76(99/435) | 20.75(50/241)[h] | 28.52(85/298) | 21.67(26/120) | 0.122 |
|     Unexplained factor % (n) | 11.29(63/558)[b] | 13.33(58/435) | 18.67(45/241) | 13.39(39/298) | 12.50(15/120) | 0.090 |
| Transplant cycle | | | | | | |
|     First cycle % (n) | 71.68(400/558)[a,b,c,d] | 36.09(157/435) | 34.85(84/241) | 47.31(141/298)[f,h] | 51.67(62/120)[g,i] | <0.001* |
|     Second cycle % (n) | 28.32(158/558)[a,b,c,d] | 63.91(278/435) | 65.15(157/241) | 52.68(157/298)[f,h] | 48.33(58/120)[g,i] | <0.001* |
| Basal hormone levels | | | | | | |
|     LH, mean (SD) (IU/L) | 5.51 ± 3.21 | 5.41 ± 3.19 | 5.32 ± 2.90 | 5.81 ± 3.50 | 5.17 ± 3.20 | 0.277 |
|     FSH, mean (SD) (IU/L) | 6.80 ± 1.87 | 6.86 ± 2.11 | 7.01 ± 2.10 | 6.84 ± 1.73 | 7.08 ± 2.24 | 0.515 |
|     E2, mean (SD) (pg/mL) | 42.63 ± 12.37 | 43.97 ± 13.74 | 44.19 ± 12.84 | 44.17 ± 12.77 | 43.69 ± 13.50 | 0.334 |
|     P, mean (SD) (ng/mL) | 0.52 ± 0.20 | 0.51 ± 0.20 | 0.52 ± 0.20 | 0.53 ± 0.20 | 0.51 ± 0.19 | 0.497 |
|     PRL, mean (SD) (mIu/L) | 12.58 ± 5.47 | 12.36 ± 5.44 | 13.02 ± 5.53 | 13.03 ± 5.47 | 11.96 ± 4.86 | 0.216 |
| Endometrial thickness on the transformation day, mean (SD) (mm) | 9.23 ± 1.47 | 9.16 ± 1.45 | 9.07 ± 1.34 | 9.07 ± 1.56 | 9.02 ± 1.42 | 0.403 |

**Notes.**

[#]Both factors were defined as more than one reason causing infertility.

*$P < 0.05$ was statistical significance. "a" represents $P$ value less than 0.05 between groups A and B, "b" represents $P$ value less than 0.05 between groups A and C, "c" represents $P$ value less than 0.05 between groups A and D, "d" represents $P$ value less than 0.05 between groups A and E, "e" represents $P$ value less than 0.05 between groups B and C, "f" represents $P$ value less than 0.05 between groups B and D, "g" represents $P$ value less than 0.05 between groups B and E, "h" represents $P$ value less than 0.05 between groups C and D, "i" represents $P$ value less than 0.05 between groups C and E, "j" represents $P$ value less than 0.05 between groups D and E.

SD, Standard deviation; LH, Luteinizing hormone; FSH, Follicle stimulating hormone; E2, Estradiol; P, Progesterone; PRL, prolactin.

B, C, and E (46.49%, $P = 0.009$; 44.74%, $P = 0.015$; 37.50%, $P < 0.001$). The proportions of male infants born in groups D (59.41%) was significantly higher than the proportions of male infants born in groups B, C, and E (46.49%, 44.74%, 37.50%, respectively; all $P < 0.001$; Table 2; Fig. 3).

## Main factors affecting live birth

The 1,652 patients included in this study were divided into a live birth group ($n = 884$) or a non-live birth group ($n = 768$) according to whether the ART led to live birth. In univariable analysis, the main factors associated with live birth were infertility duration, BMI, transplant cycle, number of embryos transferred, high-quality blastocyst transfer, and poor-quality blastocyst transfer.

A multivariable logistic regression analysis excluded poor-quality blastocyst transfer as it had a $P$ value higher than 0.05. Dominant risk factors for not having a live birth included increased infertility duration (1.270, 95% CI [1.040–1.551], $P = 0.019$), 24 $\leq$ BMI<28 (1.642, 95% CI [1.146–2.353], $P = 0.007$) and being on the second transplant cycle (1.357, 95% CI [1.101–1.672], $P = 0.004$). Double blastocyst transfer (0.528, 95% CI [0.410–0.680], $P < 0.001$) and high-quality blastocyst transfer (0.609, 95% CI [0.453–0.820], $P = 0.001$) were found to be protective factors for live birth (Table 3).

## Dominant predictors for pregnancy complications

Females with serum hCG $\geq$ 15 mlU/mL ($n = 1,274$) were divided into two groups according to the occurrence of pregnancy complications. In univariable analysis, the main factors associated with pregnancy complications were BMI, transplant cycle, number of embryos transferred, high-quality blastocyst transfer, poor-quality blastocyst transfer, and endometrial thickness.

A multivariable logistic regression analysis excluded transplant cycle, high-quality blastocyst transfer, and poor-quality blastocyst transfer as they had $P$ values higher than 0.05. Dominant risk factors for pregnancy complications were: 24 $\leq$ BMI<28 (1.646, 95% CI [1.092–2.481], $P = 0.017$), double blastocyst transfer (3.120, 95% CI [2.323–4.190], $P < 0.001$), and endometrial thickness >12 mm (2.572, 95% CI [1.295–5.109], $P = 0.013$; Table 4).

In addition, a multivariable logistic regression analysis performed on gemellary pregnancy among pregnancy complications excluded the transplant cycle as it had a $P$ value higher than 0.05. Dominant risk factors for gemellary pregnancy were 18.5 $\leq$ BMI<24 (1.611, 95% CI [1.046–2.481], $P = 0.030$) and double blastocyst transfer(59.933, 95% CI [27.298–131.58], $P < 0.001$). Poor-quality blastocyst transfer (0.656, 95% CI [0.481–0.894], $P = 0.008$) was protective against gemellary pregnancy (Table 5).

## Dominant predictors of neonatal complications

All live births ($n = 884$) were divided into two groups based on the presence or absence of neonatal complications. In univariable analysis, the main factors associated with neonatal complications were infertility duration, BMI, maternal age, transplant cycle, number of embryos transferred, poor-quality blastocyst transfer, and endometriosis.

Wu et al. (2022), *PeerJ*, DOI 10.7717/peerj.14424

**Table 2  Comparison of pregnancy outcomes and neonatal outcomes.**

| | Group A (n = 558) | Group B (n = 435) | Group C (n = 241) | Group D (n = 298) | Group E (n = 120) | P value |
|---|---|---|---|---|---|---|
| Positive rate of hCG test % (n) | 76.52(427/558) | 81.38(354/435) | 77.18(186/241) | 81.21(242/298) | 54.17(65/120)[d,g,i,j] | <0.001* |
| Clinical pregnancy rate % (n) | 67.38(376/558) | 68.74(299/435)[e] | 60.58(146/241) | 73.15(218/298)[h] | 37.50(45/120)[d,g,i,j] | <0.001* |
| Embryo implantation rate % (n) | 67.38(376/558)[a,b,c,d] | 49.77(433/870)[e] | 41.70(201/482) | 55.20(329/596)[h] | 37.50(45/120)[g,j] | <0.001* |
| Biochemical pregnancy rate % (n) | 9.14(51/558)[b,d] | 12.64(55/435) | 16.60(40/241) | 8.05(24/298)[h] | 16.67(20/120)[j] | 0.003* |
| Miscarriage rate % (n) | 21.27(80/376) | 23.41(70/299) | 28.77(42/146) | 19.72(43/218)[h] | 28.89(13/45)[j] | 0.230 |
| Ectopic pregnancy rate % (n) | 0.80(3/376) | 0.67(2/299) | 1.37(2/146) | 0.92(2/218) | 0(0/45) | 0.860 |
| Gemellary pregnancy rate % (n) | 1.60(6/376)[a,b,c] | 44.82(134/299) | 37.67(55/146)[#] | 50.92(111/218)[h] | 2.22(1/45)[g,i,j] | <0.001* |
| Preterm birth rate % (n) | 5.58(21/376)[a,b,c] | 17.06(51/299) | 17.13(24/146) | 19.26(42/218) | 6.67(3/45)[j] | <0.001* |
| 34 weeks ≤ gestational age <37 weeks | 4.52(17/376)[a,b,c] | 13.38(40/299) | 15.75(23/146) | 16.51(36/218) | 4.44(2/45)[i,j] | <0.001* |
| 28 weeks ≤ gestational age <34 weeks | 1.06(4/376) | 3.68(11/299)[e] | 0.68(1/146) | 2.75(6/218) | 2.22(1/45) | 0.103 |
| Live birth rate % (n) | 52.51(293/558)[c] | 58.62(255/435)[e] | 48.55(117/241) | 62.75(187/298 )[h] | 26.67(32/120)[d,g,i,j] | <0.001* |
| Neonatal birth weight, median(IQR) (g)[U] | 3350(3000,3650)[a,b,c] | 2750(2350,3300) | 2950(2467,3300) | 2750(2450,3300) | 3345(3000,3755)[g,i,j] | <0.001* |
| Neonatal birth age, median (IQR) (weeks)[U] | 39.57(38.71,40.34)[a,b,c] | 38.86(37.14,40.00) | 38.43(37.07,39.50) | 38.29(37.14,39.86) | 39.64(38.21,40.54)[g,i,j] | <0.001* |
| Incidence of macrosomia % (n) | 9.03(27/299)[c] | 4.68(16/342) | 3.95(6/152) | 2.95(8/271) | 15.63(5/32)[g,i,j] | 0.002* |
| Incidence of Low birth weight infants % (n) | 7.02(21/299)[a,b,c] | 27.78(95/342) | 26.32(40/152) | 27.68(75/271) | 12.50(4/32)[g,j] | <0.001* |
| 1500g ≥birth weight <2500g | 6.69(20/299)[a,b,c] | 24.27(83/342) | 26.32(40/152) | 24.35(66/271) | 6.25(2/32)[g,i,j] | <0.001* |
| Birth weight <1500g | 0.03(1/299)[a,c,d] | 3.51(12/342)[e] | 0(0/152) | 3.32(9/271)[h] | 6.25(2/32)[i] | 0.001* |
| Birth defect rate % (n) | 0.67(2/299) | 1.17(4/342) | 1.32(2/152) | 0.74(2/271) | 3.12(1/32) | 0.782 |
| Neonatal sex ratio % (n) | | | | | | 0.001* |
| Male | 56.86(170/299)[a,b,d] | 46.49(159/342) | 44.74(68/152) | 59.41(161/271)[f,h,j] | 37.50(12/32) | |
| Female | 43.14(129/299) | 53.51(183/342) | 55.26(84/152) | 40.59(110/271) | 62.50(20/32) | |
| Obstetric complications | | | | | | |
| Gestational hypertension % (n) | 0.27(1/376) | 1.34(4/299) | 1.37(2/146) | 1.38(3/218) | 2.22(1/45) | 0.384 |
| ICP % (n) | 0(0/376) | 0(0/299) | 0.68(1/146) | 0.46(1/218) | 0(0/45) | 0.348 |
| GDM % (n) | 3.19(12/376) | 2.01(6/299) | 6.85(10/146) | 2.29(5/218) | 2.22(1/45) | 0.129 |

**Notes.**

[#]In Group C, 55 multiple pregnancies included 2 monochorionic diamniotic twins, and the remaining 53 were dichorionic diamniotic twins.

*$P < 0.05$ was statistical significifance. "a" represents $P$ value less than 0.05 between groups A and B, "b" represents $P$ value less than 0.05 between groups A and C, "c" represents $P$ value less than 0.05 between groups A and D, "d" represents $P$ value less than 0.05 between groups A and E, "e" represents $P$ value less than 0.05 between groups B and C, "f" represents $P$ value less than 0.05 between groups B and D, "g" represents $P$ value less than 0.05 between groups B and E, "h" represents $P$ value less than 0.05 between groups C and D, "i" represents $P$ value less than 0.05 between groups C and E, "j" represents $P$ value less than 0.05 between groups D and E.

IQR, Interquartile range; ICP, Intrahepatic Cholestasis; GDM, Gestational Diabetes Mellitus.

[U]Kruskal-Wallis H test/groups individually tested by Mann–Whitney U-test.

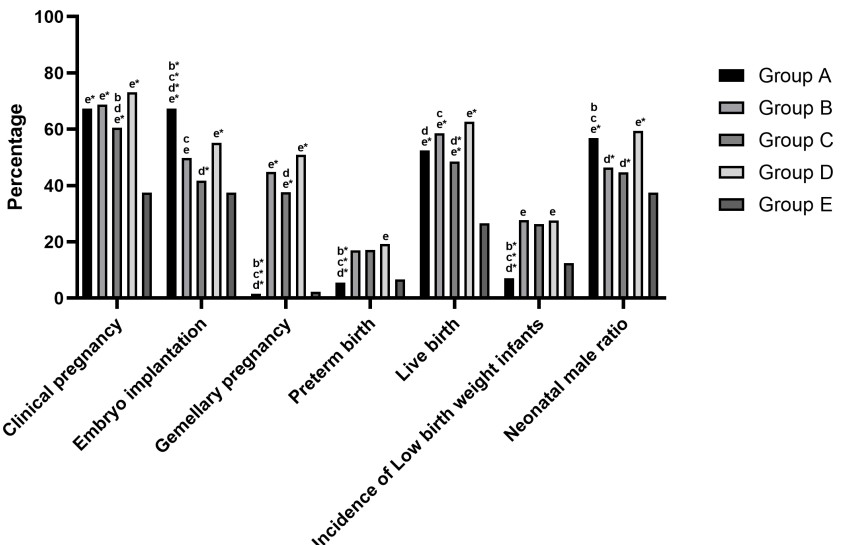

**Figure 3** **Comparison of pregnancy outcomes and neonatal outcomes between the five groups.** Note: "b" represents *P* value less than 0.05 between group B and other group. "c" represents *P* value less than 0.05 between group C and other group. "d" represents *P* value less than 0.05 between group D and other group. "e" represents *P* value less than 0.05 between group E and other group. An asterisk (*) represents *P* value less than 0.001.

A multivariable logistic regression analysis excluded maternal age, transplant cycle, and poor-quality blastocyst transfer as they had *P* values higher than 0.05. Dominant risk factors for neonatal complications included increased infertility duration (1.417, 95% CI [1.053–1.906], *P* = 0.021), 18.5 ≤BMI<24 (1.674, 95% CI [1.062–2.638], *P* = 0.026), 24 ≤BMI<28 (2.408, 95% CI [1.390–4.172], *P* = 0.002), BMI>28 (2.776, 95% CI [1.209–6.371], *P* = 0.016), double blastocyst transfer (2.230, 95% CI [1.515–3.280], *P* < 0.001), and endometriosis (3.009, 95% CI [1.250–7.247], *P* = 0.014; Table 6).

Additionally, a multivariable logistic regression analysis performed on preterm birth among neonatal complications excluded the transplant cycle and poor-quality blastocyst transfer as they had *P* values higher than 0.05. Dominant risk factors for preterm birth included 24 ≤BMI<28 (2.244, 95% CI [1.125–4.474], *P* = 0.022), double blastocyst transfer (3.840, 95% CI [2.272–6.489], *P* < 0.001), and endometriosis (3.183, 95% CI [1.259–8.048], *P* = 0.014; Table 7).

## ROC curve analysis

We further analyzed the correlations between double blastocyst transfer and high-quality blastocyst transfer with live birth, pregnancy complications, and neonatal complications using ROC curves.

We found that the ROC curves for high-quality blastocyst transfer and live birth showed a moderate correlation with an AUC of 61.20% (95% CI [0.586–0.640]) and *P* value <0.001, but high-quality blastocyst transfer did not predict pregnancy complications and neonatal complications. We also found that the ROC curves for double blastocyst transfer with gemellary pregnancy showed a stronger correlation with an AUC of 78.53%

**Table 3  Main factors affecting live birth.**

| Factors | Univariable | | Multivariable | |
|---|---|---|---|---|
| | OR (95% CI) | *P* value | OR (95% CI) | *P* value |
| **Infertility duration (year)** | | | | |
| <3 | Ref | | | |
| ≥3 | 1.321 (1.087–1.604) | 0.005[*] | 1.270 (1.040–1.551) | 0.019[*] |
| **BMI (kg/m²)** | | | | |
| <18.5 | Ref | | | |
| 18.5 ≤BMI<24 | 1.297 (0.970–1.734) | 0.080 | 1.291 (0.961–1.734) | 0.090 |
| 24 ≤BMI<28 | 1.710 (1.202–2.432) | 0.003[*] | 1.642 (1.146–2.353) | 0.007[*] |
| ≥28 | 1.658 (0.976–2.817) | 0.061 | 1.483 (0.862–2.553) | 0.154 |
| **Female age (year)** | | | | |
| < 25 | Ref | | | |
| 25–35 | 1.099 (0.750–1.610) | 0.630 | | |
| **Infertility type** | | | | |
| Primary infertility | Ref | | | |
| Secondary infertiliy | 1.204 (0.991–1.462) | 0.061 | | |
| **Transplant cycle** | | | | |
| First cycle | Ref | | | |
| Second cycle | 1.244 (1.025–1.509) | 0.027[*] | 1.357 (1.101–1.672) | 0.004[*] |
| **Number of blastocyst transfer** | | | | |
| Single | Ref | | | |
| Double | 0.684 (0.561–0.832) | <0.001[*] | 0.528 (0.410–0.680) | <0.001[*] |
| **High-quality blastocyst transfer** | | | | |
| No | Ref | | | |
| Yes | 0.532 (0.420–0.674) | <0.001[*] | 0.609 (0.453–0.820) | 0.001[*] |
| **Poor-quality blastocyst transfer** | | | | |
| No | Ref | | | |
| Yes | 1.239 (1.021–1.503) | 0.030[*] | 1.261 (0.944–1.684) | 0.117 |
| **PCOS** | | | | |
| No | Ref | | | |
| Yes | 0.945 (0.723–1.236) | 0.681 | | |
| **Endometriosis** | | | | |
| No | Ref | | | |
| Yes | 0.594 (0.294–1.202) | 0.148 | | |
| **Endometrial thickness (mm)** | | | | |
| <8 | Ref | | | |
| 8–12 | 1.041 (0.798–1.358) | 0.767 | | |
| >12 | 1.322 (0.762–2.293) | 0.321 | | |

**Notes.**
[*]*P* < 0.05 was statistical signifificance.
BMI, Body Mass Index; PCOS, Polycystic Ovary Syndrome.

(95% CI [0.760–0.811]) and *P* value <0.001. In addition, double blastocyst transfer was correlated with live birth (AUC: 61.20%, 95% CI [0.586–0.640], *P* < 0.001), pregnancy complications (AUC: 65.90%, 95% CI [0.629–0.689], *P* < 0.001), neonatal complications

**Table 4  Dominant predictors for pregnancy complications.**

| Factors | Univariable | | Multivariable | |
| --- | --- | --- | --- | --- |
| | OR (95% CI) | P value | OR (95% CI) | P value |
| **Infertility duration (year)** | | | | |
| <3 | Ref | | | |
| ≥3 | 1.163 (0.932-1.451) | 0.181 | | |
| **BMI (kg/m²)** | | | | |
| <18.5 | Ref | | | |
| 18.5 ≤BMI<24 | 1.371 (0.999–1.882) | 0.051 | 1.339 (0.962–1.863) | 0.084 |
| 24 ≤BMI<28 | 1.748 (1.181–2.588) | 0.005* | 1.646 (1.092–2.481) | 0.017* |
| ≥28 | 1.453 (0.801–2.636) | 0.219 | 1.634 (0.876–3.048) | 0.123 |
| **Female age (year)** | | | | |
| < 25 | Ref | | | |
| 25–35 | 1.159 (0.745–1.805) | 0.513 | | |
| **Infertility type** | | | | |
| Primary infertility | Ref | | | |
| Secondary infertiliy | 1.164 (0.933–1.452) | 0.177 | | |
| **Transplant cycle** | | | | |
| First cycle | Ref | | | |
| Second cycle | 1.398 (1.120–1.745) | 0.003* | 1.018 (0.798–1.298) | 0.887 |
| **Number of blastocyst transfer** | | | | |
| Single | Ref | | | |
| Double | 3.085 (2.441–3.899) | <0.001* | 3.120 (2.323–4.190) | <0.001* |
| **High-quality blastocyst transfer** | | | | |
| No | Ref | | | |
| Yes | 0.626 (0.471–0.832) | 0.001* | 0.706 (0.496–1.006) | 0.054 |
| **Poor-quality blastocyst transfer** | | | | |
| No | Ref | | | |
| Yes | 1.962 (1.568–2.456) | <0.001* | 0.936 (0.674–1.300) | 0.694 |
| **PCOS** | | | | |
| No | Ref | | | |
| Yes | 0.975 (0.723–1.315) | 0.868 | | |
| **Endometriosis** | | | | |
| No | Ref | | | |
| Yes | 1.181 (0.559–2.493) | 0.663 | | |
| **Endometrial thickness (mm)** | | | | |
| <8 | Ref | | | |
| 8–12 | 1.162 (0.856–1.576) | 0.336 | 1.214 (0.882–1.672) | 0.234 |
| >12 | 2.457 (1.270–4.753) | 0.008* | 2.572 (1.295–5.109) | 0.013* |

Notes.
*P < 0.05 was statistical siginificance.
BMI, Body Mass Index; PCOS, Polycystic Ovary Syndrome.

**Table 5  Dominant predictors for gemellary pregnancy.**

| Factors | Univariable | | Multivariable | |
|---|---|---|---|---|
| | OR (95% CI) | *P* value | OR (95% CI) | *P* value |
| **Infertility duration (year)** | | | | |
| <3 | Ref | | | |
| ≥3 | 1.019 (0.787–1.317) | 0.889 | | |
| **BMI (kg/m2)** | | | | |
| <18.5 | Ref | | | |
| 18.5 ≤BMI<24 | 1.556 (1.048–2.310) | 0.028* | 1.611 (1.046–2.481) | 0.030* |
| 24 ≤BMI<28 | 1.187 (0.732–1.925) | 0.487 | 1.097 (0.649–1.855) | 0.729 |
| ≥28 | 0.698 (0.304–1.603) | 0.397 | 0.837 (0.339–2.069) | 0.700 |
| **Female age (year)** | | | | |
| <25 | Ref | | | |
| 25–35 | 1.376 (0.785–2.409) | 0.256 | | |
| **Infertility type** | | | | |
| Primary infertility | Ref | | | |
| Secondary infertiliy | 1.100 (0.850–1.424) | 0.467 | | |
| **Transplant cycle** | | | | |
| First cycle | Ref | | | |
| Second cycle | 1.393 (1.076–1.802) | 0.012* | 0.775 (0.578–1.038) | 0.087 |
| **Number of blastocyst transfer** | | | | |
| Single | Ref | | | |
| Double | 43.124 (20.169–92.203) | <0.001* | 59.933 (27.298–131.580) | <0.001* |
| **High-quality blastocyst transfer** | | | | |
| No | Ref | | | |
| Yes | 1.132 (0.814–1.574) | 0.460 | | |
| **Poor-quality blastocyst transfer** | | | | |
| No | Ref | | | |
| Yes | 2.160 (1.660–2.810) | <0.001* | 0.656 (0.481–0.894) | 0.008* |
| **PCOS** | | | | |
| No | Ref | | | |
| Yes | 0.942 (0.662–1.339) | 0.738 | | |
| **Endometriosis** | | | | |
| No | Ref | | | |
| Yes | 1.959 (0.915–4.195) | 0.083 | | |
| **Endometrial thickness (mm)** | | | | |
| <8 | Ref | | | |
| 8–12 | 1.075 (0.749–1.544) | 0.695 | | |
| >12 | 1.333 (0.673–2.642) | 0.410 | | |

**Notes.**
*$P < 0.05$ was statistical siginificance.
BMI, Body Mass Index; PCOS, Polycystic Ovary Syndrome.

**Table 6  Dominant predictors for neonatal complications.**

| Factors | Univariable | | Multivariable | |
|---|---|---|---|---|
| | OR (95% CI) | P value | OR (95% CI) | P value |
| **Infertility duration (year)** | | | | |
| <3 | Ref | | | |
| ≥3 | 1.368 (1.028–1.821) | 0.032[*] | 1.417 (1.053–1.906) | 0.021[*] |
| **BMI (kg/m2)** | | | | |
| <18.5 | Ref | | | |
| 18.5 ≤BMI<24 | 1.738 (1.113–2.713) | 0.015[*] | 1.674 (1.062–2.638) | 0.026[*] |
| 24 ≤BMI<28 | 2.482 (1.452–4.242) | 0.001[*] | 2.408 (1.390–4.172) | 0.002[*] |
| ≥28 | 2.241 (1.009–4.980) | 0.048[*] | 2.776 (1.209–6.371) | 0.016[*] |
| **Female age (year)** | | | | |
| <25 | Ref | | | |
| 25–35 | 2.018 (1.030–3.954) | 0.041[*] | 1.966 (0.988–3.910) | 0.054 |
| **Infertility type** | | | | |
| Primary infertility | Ref | | | |
| Secondary infertiliy | 1.074 (0.808–1.428) | 0.623 | | |
| **Transplant cycle** | | | | |
| First cycle | Ref | | | |
| Second cycle | 1.593 (1.197–2.121) | 0.001[*] | 1.316 (0.965–1.794) | 0.083 |
| **Number of blastocyst transfer** | | | | |
| Single | Ref | | | |
| Double | 2.240 (1.630–3.077) | <0.001[*] | 2.230 (1.515–3.280) | <0.001[*] |
| **High-quality blastocyst transfer** | | | | |
| No | Ref | | | |
| Yes | 0.742 (0.513–1.075) | 0.115 | | |
| **Poor-quality blastocyst transfer** | | | | |
| No | Ref | | | |
| Yes | 1.524 (1.145–2.028) | 0.004[*] | 0.920 (0.649–1.305) | 0.641 |
| **PCOS** | | | | |
| No | Ref | | | |
| Yes | 1.094 (0.744–1.608) | 0.649 | | |
| **Endometriosis** | | | | |
| No | Ref | | | |
| Yes | 2.467 (1.075–5.662) | 0.033[*] | 3.009 (1.250–7.247) | 0.014[*] |
| **Endometrial thickness (mm)** | | | | |
| <8 | Ref | | | |
| 8–12 | 1.063 (0.719–1.571) | 0.760 | | |
| >12 | 1.163 (0.502–2.691) | 0.725 | | |

**Notes.**
[*]$P < 0.05$ was statistical signififance.
BMI, Body Mass Index; PCOS, Polycystic Ovary Syndrome.

**Table 7  Dominant predictors for premature birth.**

| Factors | Univariable | | Multivariable | |
|---|---|---|---|---|
| | OR (95% CI) | *P* value | OR (95% CI) | *P* value |
| **Infertility duration (year)** | | | | |
| <3 | Ref | | | |
| ≥3 | 1.268 (0.890–1.807)) | 0.189 | | |
| **BMI (kg/m²)** | | | | |
| <18.5 | Ref | | | |
| 18.5 ≤BMI<24 | 1.716 (0.963–3.058) | 0.067 | 1.615 (0.897–2.909) | 0.110 |
| 24 ≤BMI<28 | 2.283 (1.162–4.486) | 0.017* | 2.244 (1.125–4.474) | 0.022* |
| ≥28 | 1.710 (0.611–4.788) | 0.307 | 2.359 (0.814–6.834) | 0.114 |
| **Female age (year)** | | | | |
| <25 | Ref | | | |
| 25–35 | 2.238 (0.879–5.697) | 0.091 | | |
| **Infertility type** | | | | |
| Primary infertility | Ref | | | |
| Secondary infertiliy | 1.156 (0.812–1.646) | 0.422 | | |
| **Transplant cycle** | | | | |
| First cycle | Ref | | | |
| Second cycle | 1.519 (1.066–2.164) | 0.021* | 1.139 (0.779–1.665) | 0.502 |
| **Number of blastocyst transfer** | | | | |
| Single | Ref | | | |
| Double | 3.612 (2.278–5.727) | <0.001* | 3.840 (2.272–6.489) | <0.001* |
| **High-quality blastocyst transfer** | | | | |
| No | Ref | | | |
| Yes | 0.751 (0.481–1.174) | 0.209 | | |
| **Poor-quality blastocyst transfer** | | | | |
| No | Ref | | | |
| Yes | 1.622 (1.138–2.312) | 0.008* | 0.855 (0.566–1.289) | 0.454 |
| **PCOS** | | | | |
| No | Ref | | | |
| Yes | 1.087 (0.677–1.745) | 0.730 | | |
| **Endometriosis** | | | | |
| No | Ref | | | |
| Yes | 2.723 (1.133–6.545) | 0.025* | 3.183 (1.259–8.048) | 0.014* |
| **Endometrial thickness (mm)** | | | | |
| <8 | Ref | | | |
| 8–12 | 1.128 (0.688–1.850) | 0.633 | | |
| >12 | 1.375 (0.504–3.750) | 0.534 | | |

**Notes.**
*$P < 0.05$ was statistical siginificance.
BMI, Body Mass Index; PCOS, Polycystic Ovary Syndrome.

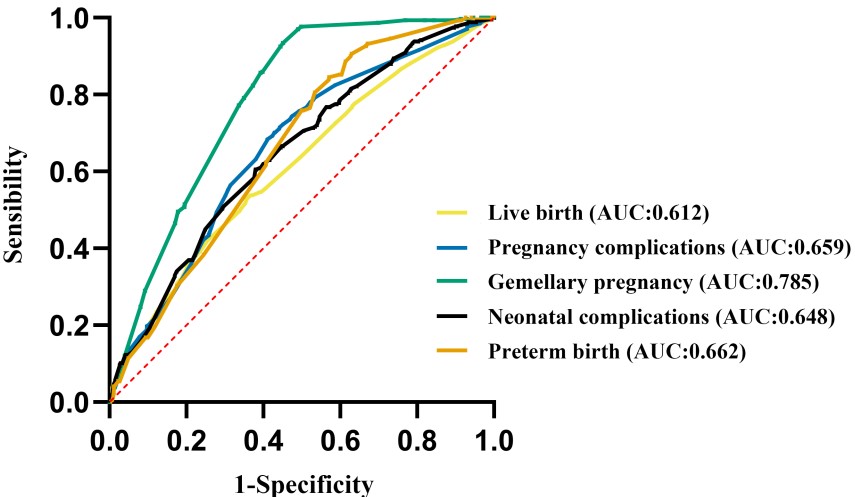

**Figure 4   Receiver operating characteristic (ROC) curves analysis.** ROC curves were used to show whether double embryo transfer and high-quality blastocyst transfer could be used to predict the accuracy of live birth, pregnancy complications, gemellary pregnancy, neonatal complications and preterm birth.

(AUC: 64.80%, 95% CI [0.610–0.686], *P* < 0.001), and preterm birth (AUC: 66.20%, 95% CI [0.620–0.705], *P* < 0.001; Fig. 4).

## DISCUSSION

To improve pregnancy outcomes, conventional ART techniques often adopt double blastocyst transfer, increasing the risk of gemellary pregnancy. Gemellary pregnancy is high risk and can seriously threaten the health of both the mother and the baby. Embryo number is significantly correlated with pregnancy outcomes. Embryo reduction and reducing the number of embryos transferred both stop gemellary pregnancy. However, embryo reduction is a remedial measure, while reducing the number of embryos transferred is a preventive measure. Studies have shown that a frozen-thawed single blastocyst transfer can effectively prevent gemellary pregnancy and reduce pregnancy and neonatal complications (*Tannus et al., 2017*). However, because there is no consensus on SET, there is concern that SET may affect the pregnancy outcomes of a single transfer cycle. Fully implementing SET remains controversial in most countries, including China and the United States (*Adashi & Gleicher, 2017*). Based on the reduction of gemellary pregnancy seen in SET, it is a reasonable transplantation strategy.

In order to investigate the effect of the number and quality of transferred blastocysts in FET on pregnancy outcomes and to formulate the optimal FET strategy for blastocyst transfer, patients undergoing frozen-thawed blastocyst transfer were divided into five groups according to the number and quality of the transferred blastocysts. The results showed that group A had a clinical pregnancy rate similar to other groups (except for group E), without increasing the risk of adverse pregnancies such as miscarriage, biochemical pregnancy, ectopic pregnancy, and other obstetric-related complications. Although the live birth rate in group D was higher than in group A, the gemellary pregnancy rate and

preterm birth rate in groups B, C, and D, were significantly higher than in group A. By conducting a multi-factor logistic regression analysis, we found that double blastocyst transfer (OR:0.528, 95% CI [0.410–0.680]) and high-quality blastocyst transfer (OR:0.609, 95% CI [0.453–0.820]) were protective factors for live birth, but double blastocyst transfer was also a risk factor for pregnancy complications (OR:3.120, 95% CI [2.323–4.190]) and neonatal complications (OR:2.230, 95% CI [1.515–3.280]). For example, double blastocyst transfer was associated with a nearly 60-fold increase in the probability of gemellary pregnancy (OR:59.933, 95% CI [27.298–131.58]) and a 3.84-fold increase in the incidence of preterm birth (OR:3.840, 95% CI [2.272–6.489]) compared to single blastocyst transfer. *Jacobs, Klonoff-Cohen & Garzo (2018)* found that single blastocyst transfer has a clinical pregnancy rate similar to double blastocyst transfer. It is well known that premature infants, especially low birth weight infants, have immature organ development. The pregnancy complications caused by gemellary pregnancy also significantly increase the risk factors for perinatal infants. These complications also add to obstetricians' workload and increase patients' financial burden and mental stress (*Valenzuela-Alcaraz et al., 2018*). It is universally accepted that while double blastocyst transfer has a high live birth rate, this also leads to a high rate of gemellary pregnancy and related complications. This study highlights that the choice of transplantation strategy cannot rely solely on superiority analyses alone.

According to a recent meta-analysis (*Ma et al., 2022*), for young patients, SET has a lower live birth rate and a lower rate of gemellary pregnancy than DET. Considering the risks of gemellary pregnancy, we believe that SET may be more beneficial. Over the past two decades, evidence has demonstrated that the transfer of morphologically high-quality blastocysts has higher implantation and pregnancy rates than poor-quality blastocyst transfer (*Balaban et al., 2000*; *Gardner et al., 2000*). This means that patients with a high-quality embryo are excellent candidates for a single blastocyst transfer (*Heitmann et al., 2013*). Our study also found that the live birth rate of poor-quality blastocyst transfer in SET was much lower than that of high-quality blastocyst transfer. This study also suggests that high-quality single blastocyst transfer can achieve stable clinical pregnancy and live birth rate without increasing adverse pregnancy complications, such as miscarriage. It also shows that high-quality single blastocyst transfer significantly reduces gemellary pregnancy rate, preterm birth, and low birth weights, achieving a qualitative leap in maternal and infant safety. After weighing the pros and cons of live birth with pregnancy and neonatal complications, the authors believe that high-quality single blastocyst transfer is the optimal FET strategy, which is the same conclusion reached by *Chen et al. (2020)*.

Embryo implantation rate refers to the ratio of implanted embryos to transferred embryos. In frozen-thawed blastocyst transfer, the implantation rate of elective DET is significantly lower than that of SET (30.9% *vs.* 52.5%; *Monteleone et al., 2016*). A segmentation study on embryo quality showed that the embryo implantation rate of the single high-quality embryo group was the highest and significantly higher than that of the high-quality embryo plus the poor-quality embryo group and the double high-quality embryo group. The researchers believe that poor embryo morphological development can significantly reduce the implantation rate of high-quality embryos (*Wintner et al., 2017*; *El-Danasouri et al., 2016*). There is also evidence that an embryo can send signals to

the endometrium, which has a mechanism through which its receptivity and selectivity can be continuously re-balanced. The lumen in the epithelial tissue likely transmits and amplifies signals from competent embryos, making the lower decidual layer more likely to embrace invasion, increasing the probability of a successful pregnancy. But in the case of a poor-quality embryo, the network-supporting decidua is inactivated, possibly negatively affecting endometrial receptivity (*Macklon & Brosens, 2014*). This study showed that the embryo implantation rate of group A was the highest and was significantly higher than the other four groups, while group E was the lowest, suggesting that double blastocyst transfer and poor-quality blastocyst transfer can reduce embryo utilization. *Wang et al. (2020)* found that poor-quality embryos would not adversely impact the implantation potential of co-transplanted high-quality embryos, but did increase gemellary pregnancy rate. Some scholars (*Zhu et al., 2020*; *Hill et al., 2020*) conclude that the addition of a poor-quality embryo does not adversely affect high-quality blastocysts, and even may slightly increase the live birth rate, but this is at the expense of significantly increased gemellary pregnancy rate. Regardless of the impact of poor-quality embryos on embryo implantation, DET always increases the risk of gemellary pregnancy. Therefore, simultaneously transferring a high-quality blastocyst with a poor-quality blastocyst is not recommended, nor is transferring a single poor-quality blastocyst.

It is worth noting that although SET significantly reduces gemellary pregnancy risk, it does not entirely remove the risk of gemellary pregnancy, especially for monozygotic twins (MZT). In this study, six cases of MZT occurred in group A (an incidence of 1.6%) and one MZT occurred in group E (an incidence of 2.2%). Some research indicates that blastocyst transfers can increase the rate of MZT by 4.25 times compared with natural pregnancy (1.7% *vs.* 0.4%; *Nakasuji et al., 2014*). This may be because the consecutive exposure to and treatment of zona pellucida during blastocyst transfer can stimulate the division of inner cell mass, leading to an increased MZT rate (*Yuri Shibuya & Kyono, 2012*). *Corner (1955)* developed a classical theory on embryo division and on the development time of gemellary pregnancy: within 3 days after fertilization, double chorionic double amnion (DC-DA) pregnancy is formed; within 4 to 8 days, single chorionic double amnion (MCDA) gemellary pregnancy is formed; within 9 to 12 days, single chorionic single amnion (MC-MA) gemellary pregnancy is formed; after 12 days, conjoined twins are formed. This theory is now recognized as the standard and has been quoted in books and articles. As Herranz writes, "fifteen years after its publication, the model became standard wisdom" (*Herranz, 2015*). According to this classic theory, no DC-DA pregnancy will occur after blastocyst transfer. However, in recent years, it has been reported that during IVF treatment, single blastocyst transfer can, indeed, leads to a double chorionic monozygotic twin pregnancy (*Li, Shen & Sun, 2020*; *Sundaram, Ribeiro & Noel, 2018*). Unfortunately, the above studies just include only downregulated FET, with a confirmed lack of mid-cycle ovulation preventing the possibility of dizygotic DC-DA gestations. Additionally, none of the four resulting infants had a confirmatory DNA analysis due to cost. The numbers in these studies are also too small to perform statistical analyses. Our study showed one DC-DA pregnancy in group A. As in the above studies (*Li, Shen & Sun, 2020*; *Sundaram, Ribeiro & Noel, 2018*), ovulation during intimal preparation was confirmed by B-ultrasound but was

not confirmed by DNA examination. In this case, the DC-DA pregnancy indeed occurred after blastocyst transfer. Existing research suggests that this phenomenon is caused by multiple factors, and no one definitive factor has yet been identified (*Li, Shen & Sun, 2020*; *Sundaram, Ribeiro & Noel, 2018*), but this phenomenon challenges the standard theory. More research on the mechanism of monozygotic DC-DA gestations must be done to help reduce the increased risk of monozygotic multiples associated with IVF technologies.

Neonatal outcomes also differed between the groups. Group A had significantly higher birth weights and gestational ages than the three double blastocyst transfer groups. Group A also had the lowest incidence of low birth weight, and group D had the lowest incidence of macrosomia. Group A had the lowest incidence of low birth weight because that double blastocyst transfer significantly increased the gemellary pregnancy rate. Gemellary pregnancy has significant risks including increased morbidity of newborns, premature delivery, low birth weight, and very low birth weight (*Kang et al., 2012*). The aim of ART is the birth of a single, full-term, healthy infant (*Ferraretti et al., 2013*), further indicating that reducing ART gemellary pregnancy rate should be a clinical goal. Interestingly, the proportions of males born in groups A and D were significantly higher, and significantly exceeded the proportion of males born in the overall population under natural pregnancy (103-110) reported by the World Health Organization (*Chang et al., 2009*). Several studies (*Lou et al., 2020*; *Hu et al., 2021*) also suggest that high-quality blastocysts are more likely to lead to male infants than poor-quality blastocysts. *Roos Kulmann et al. (2021)* performed biopsies on 1,254 embryos from 466 PGT-A patients, further confirming that male embryos possibly have higher TE grades. This might be because the development of male embryos is different from that of female embryos from the cleavage stage to the blastocyst stage; the embryos carrying male genetic material generally have more cells and divide faster. Therefore, male embryos are more likely to be selected as high-quality blastocysts for transfer (*Maalouf et al., 2014*). Notably, the group B did not have the same increase in the proportion of males born seen in the other high-quality blastocyst groups. This may be because there is a competitive relationship in the early stage of embryo implantation, and embryo quality may affect embryo implantation, resulting in differences in the proportions of males and females born. Currently, the number of infants born from high-quality blastocyst transfer makes up a tiny proportion of the overall population, so it does not currently affect the infant sex ratio. Nevertheless, it is unclear what impact high-quality blastocyst transfer will have on the future population composition as ART continues to develop and as the number of infants born through this technique increases. In recent years, few studies have been conducted on the sex ratio of ART infants, maybe because this is a sensitive topic or it requires a large sample size to verify the effects. Therefore, to study whether high-quality blastocyst transfer leads to differences in infant sex ratios, we need to design extensive, rigorous, multi-center clinical studies. Long-term follow-up epidemiological investigations may also be essential.

This study has three major limitations, which we expect will be solved in future studies. First, our study population only included young women and excluded women of advanced age, so it needs to be double blastocyst transferermined whether frozen-thawed high-quality single blastocyst transfer is also suitable for women of advanced age. Second, we

only investigated the pregnancy outcomes of FET fertility treatments. No analysis was done on the cumulative pregnancy rate, and the intellectual and physical development of the newborns after birth was not tracked. Finally, this was a single-center retrospective study. Large, prospective, multi-centered, randomized controlled trials that strictly control for exogenous variables are required to confirm the advantages of high-quality single blastocyst transfers in the FET cycle for treating infertility.

## CONCLUSIONS

High-quality single blastocyst transfers in the FET cycle have a high live birth rate and implantation rate, significantly lower gemellary pregnancy rate, and lower rates of premature or low birth weight infants. Maternal and infant outcomes are also significantly improved with this method. According to a multiple-factor analysis, double blastocyst transfer and high-quality blastocyst transfer were protective factors for live birth. However, double blastocyst transfer was also a risk factor for pregnancy and neonatal complications. After weighing the pros and cons of live birth with pregnancy and neonatal complications, we conclude that high-quality single blastocyst transfer remains the optimal FET strategy, but it should be noted that high-quality single blastocyst transfer does not entirely remove the risk of gemellary pregnancy, so the possibility of MZT should be considered. This study even included a DC-DA case, which contradicts the current understanding of gemellary pregnancy progression and is worthy of further confirmation. In addition, the proportion of males born from high-quality single blastocyst transfer was notably higher than in other transfer techniques. This possible impact should be considered in future studies. This study is of great clinical significance for blastocyst selection in the FET cycle and the effective reduction of ART gemellary pregnancy rate.

**Abbreviations**

| | |
|---|---|
| **FET** | Frozen-thawed embryo transfer |
| **SET** | Single embryo transfer |
| **DET** | Double embryo transfer |
| **MZT** | Monozygotic twins |
| **ART** | Assisted Reproductive Technology |
| **EMs** | Endometriosis |
| **PCOS** | Polycystic ovary syndrome |
| **HRT** | Hormone replacement therapy |
| **FSH** | Follicle-stimulating hormone |
| **LH** | Luteinizing hormone |
| **E2** | Estrogen-2 |
| **P** | Progesterone |
| **PRL** | Prolactin |
| **hCG** | Human chorionic gonadotropin |
| **ICM** | Inner cell mass |
| **TE** | Trophectoderm |
| **BMI** | Body mass index |

| GDM | Gestational Diabetes Mellitus |
| ICP | Intrahepatic Cholestasis |
| IVF-ET | In vitro fertilization and embryo transfer |
| DC-DA | Double chorionic double amnion |
| MCDA | Single chorionic double amnion |
| MC-MA | Single chorionic single amnion |

## ACKNOWLEDGEMENTS

The authors thank everyone for participating in the clinical and laboratory work.

### Funding

This study was funded by the Wenzhou City Key Innovation Team of Reproductive Genetics Grant, Zhejiang, China (No. C20170007). The funders had no role in study design, data collection and analysis, decision to publish, or preparation of the manuscript.

### Grant Disclosures

The following grant information was disclosed by the authors:
Wenzhou City Key Innovation Team of Reproductive Genetics Grant, Zhejiang, China: C20170007.

### Competing Interests

The authors declare there are no competing interests.

### Author Contributions

- Yanhong Wu conceived and designed the experiments, performed the experiments, prepared figures and/or tables, authored or reviewed drafts of the article, and approved the final draft.
- Xiaosheng Lu performed the experiments, prepared figures and/or tables, and approved the final draft.
- Yanghua Fu analyzed the data, authored or reviewed drafts of the article, and approved the final draft.
- Junzhao Zhao conceived and designed the experiments, authored or reviewed drafts of the article, and approved the final draft.
- Liangliang Ma conceived and designed the experiments, analyzed the data, prepared figures and/or tables, and approved the final draft.

### Human Ethics

The following information was supplied relating to ethical approvals (i.e., approving body and any reference numbers):

The Ethics Committee (Institutional Review Board) of the Second Affiliated Hospital and Yuying Children's Hospital of Wenzhou Medical University approved the study (2021-K-74-02).

## Data Availability

The raw measurements are available as a Supplemental File.

## Supplemental Information

Supplemental information for this article can be found online at http://dx.doi.org/10.7717/peerj.14424#supplemental-information.

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
