# Peer review of "Comparison of frozen-thawed embryo transfer strategies for the treatment of infertility in young women: a retrospective study"

_PeerJ, doi:10.7717/peerj.14424_

## Round 0.1 · original submission · Major Revisions

· Academic Editor

Major Revisions

Thank you for your submission. Based on the reviewers' comments, I suggest major revisions for this manuscript. Please respond to reviewers' comments. Below are some of my additional comments.

1. In the results section, please include the actual p-value rather than simply annotating them into either p <0.05 or p>0.05.

2. Have authors assessed the collinearity among all significant variables from the univariate analysis before fitting them to the multivariate analysis?

3. Please avoid using any causal language in the conclusion by misinterpreting the correlation as causation.

4. After addressing reviewers' comments, I highly recommend seeking professional help to ensure the revised manuscript is free from grammatical errors.

Reviewer 1 ·

Basic reporting

1. When interpreting the results, the term “significantly different” was used all over the article without further explanation (e.g., different in what direction, magnitude of difference, etc.). More detailed interpretations are needed.
2. The interpretation of OR (e.g., in the abstract and results sections) was inaccurate and should be corrected.
3. A “satisfactory pregnancy rate/outcome” was mentioned in conclusion. What outcome/rate is considered as “satisfactory” and what are the criteria? Evidence to supporting such criteria/decision is needed.
4. In the introduction section, evidence should be provided to support “Although Chinese experts have achieved a consensus on the number of embryo transfer in 2018, due to insufficient cognition of SET, especially the FET cycle, the implementation of SET in China is far inferior to that in European and American.”

Experimental design

1. The inclusion criterion “D5 blastocyst transplanted” was not defined in the research objects section.
2. The exclusion criteria should be listed in full instead of using “etc.”.
3. “A total of 1532 patients were involved and were divided into four groups according to the number and quality of transferred blastocysts” – what are the criteria of assignment?
4. What is “transplantable” blastocyst and the difference from the “high-quality” blastocyst?
5. For the criteria of blastocyst evaluation, is the score the higher the better? Please clarify.
6. What is the meaning of “different ICM and TE scores being C at the same time”? Please clarify.
7. Different testing strategies were implemented for statistical analysis according to the data distributions (i.e., normally distributed or not). However, which part of the data was treated as normally distributed and which was not, as well as the support for such classification was missing.
8. The article states that “Considering the risks brought by gemellary pregnancy, such as complications, we believe that single embryo transfer may be more beneficial.”, which suggests a fair comparison by adding a group of transplantable single blastocyst transfer before concluding that a high-quality single blastocyst transfer is “optimal”.

Validity of the findings

1. For the conclusion, the high-quality single blastocyst transfer strategy can only be “optimal” for females under 35 among the four strategies being studied, which needs to be clarified.
2. In line 271 “This study showed that double embryo transfer could not significantly improve embryo implantation rate; the embryo implantation rate of group A was the highest, and that of group C was the lowest, suggesting that double embryo transfer can reduce embryo utilization.” – the comparison between group A and C was not fair because of the different types of blastocyst used in the transfer.
3. In the discussion, “Therefore, after weighing pros and cons, we hold that single blastocyst transfer is the optimal transfer strategy” while the disadvantage of single blastocyst transfer was not clearly stated before making this conclusion.
4. The footnote in Table 2 was not correctly formatted.
5. Predictors like transplant cycle(%) and cause of infertility (%) had multiple categories but with only one p-value (e.g., Table 1), where interpretations and more information are needed.

Additional comments

No comments

Reviewer 2 ·

Basic reporting

Please check the word spelling and grammar. The manuscript also needs to be carefully edited by professional English speaker experts.

Experimental design

Ddding a flowchart of the study maybe better.

Validity of the findings

Table 3 should describe the OR (95% CI). The paper (DOI: 10.3389/fcvm.2022.860600) maybe an example.

How ahout the accuracy (e.g C-index, AUC or calibration plot) of the important variables between the live birth group and non-live birth group?

Annotated reviews are not available for download in order to protect the identity of reviewers who chose to remain anonymous.

Reviewer 3 ·

Basic reporting

1. There are many gramma and format errors. To name only a few:
1) Page 4, in Methods and line 17, there is a period between “Medical University” and “from”.
2) Page 4, in Results, “The proportion of male offspring in group A and D was significantly higher than group B and C (P < 0.05).” should be “The proportions of male offspring in group A and D were significantly higher than group B and C (P < 0.05).
3) Line 52, a powerful means should be a powerful mean or method
4) Line 61, Japan, and Australia should be Japanese, and Australian scientist (doctors)
5) Line 100, “take 100 mg…” should be “patients take 100 mg…”
6) Line 102 and 103, there should be a space before and after the parenthesis.
2. There are no figures displayed in this paper. Although figures are not required in the journal review criterion, it would be much better If some figures of results can be displayed. For example, besides Table 1-3, author can also show us the bar chart of some variables for 4 groups. It is much clearer to see difference among variables from the figures.
3. For the raw data, there is a serious problem. The raw data is incomplete. In the peerj-74699-raw_data.xlsx, data in group C and D are same, although they look like different superficially. The only difference is the order. Both groups have 298 patients. IDs in two groups should be different, but they share same IDs.

Experimental design

I suggest using a table to describe statistical methods (T-test, ANOVA, etc.) the authors used to analyze the variables. For example, author may use ANOVA to analyze the age variable, or multivariate logistic regression to analyze specific variables. Thus, it can be much clearer which statistical method you use for each variable. The table can be input in Appendix.

Validity of the findings

no comments

Additional comments

no comments

---

## Round 0.2 · Minor Revisions

· Academic Editor

Minor Revisions

Please take a look at the comments from the reviewers. In addition to that, I also have a few editorial comments.

1. In Figure 1, could you add the term "excluding" before "Cycles of b-hCG<15" and "Non-live birth" to make it clear that we are excluding these many subjects?

2. Please include the p-value of the comparison to Figure 3.

3. Is there a way that you could include the ROC value in the legend (or near the curves) for Figure 4?

4. The thousands separator (comma) was not used consistently throughout the manuscript. Please have it revised.

Reviewer 1 ·

Basic reporting

1. In the introduction section "These high rates indicate that the SET implementation in China is far inferior to SET adoption rates in Europe and the United States." Why these are high rates? Is there any reference? How could these high rates indicate inferior? Do you have evidence indicating the rates in Europe and US?
2. In the research objects section, D5, D7,... still not defined.

Experimental design

No comment.

Validity of the findings

No comment.

Reviewer 2 ·

Basic reporting

The language and references are professional English used and unambiguous. The article structure, figures, and tables are clear.

Experimental design

The experimental design is reasonable and understandable.

Validity of the findings

The manuscript is meaningful for clinical treatment.

Additional comments

Please confirm the label of the x-axis and y-axis of figure 4, it's better for the authors to figure out the meaning of ROC cure. the x-axis should be "1-Specificity".

Reviewer 3 ·

Basic reporting

The second version is much better than the first one in terms of English writing. More tables and figures are added to illustrate the results and make us correctly understand more details. All raw data are complete. I have no additional comments on modification or improvements.

Experimental design

no comments

Validity of the findings

no comments.

Additional comments

no comments

---

## Round 0.3 · accepted · Accept

· Academic Editor

Accept

Thank you for addressing all the comments.

Reviewer 1 ·

Basic reporting

No comment

Experimental design

No comment

Validity of the findings

No comment

Reviewer 2 ·

Basic reporting

no comments

Experimental design

no comments

Validity of the findings

no comments